# Identification and Characterization of Dmct: A Cation Transporter in *Yarrowia lipolytica* Involved in Metal Tolerance

**DOI:** 10.3390/jof9060600

**Published:** 2023-05-23

**Authors:** Katia Jamileth González-Lozano, Elva Teresa Aréchiga-Carvajal, Zacarías Jiménez-Salas, Debany Marlen Valdez-Rodríguez, Claudia Geraldine León-Ramírez, José Ruiz-Herrera, Juan Manuel Adame-Rodríguez, Manuel López-Cabanillas-Lomelí, Eduardo Campos-Góngora

**Affiliations:** 1Universidad Autónoma de Nuevo León, Facultad de Ciencias Biológicas, Departamento de Microbiología, LMYF, Unidad de Manipulación Genética, Monterrey CP 66455, Nuevo León, Mexico; 2Universidad Autónoma de Nuevo León, Centro de Investigación en Nutrición y Salud Pública, Monterrey CP 64460, Nuevo León, Mexico; 3Centro de Investigación y de Estudios Avanzados del Instituto Politécnico Nacional, Unidad Irapuato, Departamento de Ingeniería Genética, Irapuato CP 36824, Guanajuato, Mexico

**Keywords:** cation efflux, divalent metal cation transporter, *DMCT* gene, *Yarrowia lipolytica*, YALI0F19734p, YALI0F19734g, *Yl*-Dmct hypothetical protein

## Abstract

*Yarrowia lipolytica* is a dimorphic fungus used as a model organism to investigate diverse biotechnological and biological processes, such as cell differentiation, heterologous protein production, and bioremediation strategies. However, little is known about the biological processes responsible for cation concentration homeostasis. Metals play pivotal roles in critical biochemical processes, and some are toxic at unbalanced intracellular concentrations. Membrane transport proteins control intracellular cation concentrations. Analysis of the *Y. lipolytica* genome revealed a characteristic functional domain of the cation efflux protein family, i.e., YALI0F19734g, which encodes YALI0F19734p (a putative *Yl*-Dmct protein), which is related to divalent metal cation tolerance. We report the in silico analysis of the putative *Yl*-Dmct protein’s characteristics and the phenotypic response to divalent cations (Ca^2+^, Cu^2+^, Fe^2+^, and Zn^2+^) in the presence of mutant strains, Δ*dmct* and R*dmct*, constructed by deletion and reinsertion of the *DMCT* gene, respectively. The absence of the *Yl*-Dmct protein induces cellular and growth rate changes, as well as dimorphism differences, when calcium, copper, iron, and zinc are added to the cultured medium. Interestingly, the parental and mutant strains were able to internalize the ions. Our results suggest that the protein encoded by the *DMCT* gene is involved in cell development and cation homeostasis in *Y. lipolytica*.

## 1. Introduction

Metals play pivotal roles in critical biochemical processes for cellular survival [1]; however, some can become toxic when the intracellular concentration is not balanced [2,3,4]. Intracellular cation concentration stability is facilitated by transmembrane proteins that permit an adequate ionic balance for the cell to realize all vital functions [3]. These transporters drive metal translocation across the membrane, generally against the electrochemical gradient, conferring tolerance on metal cations [5,6,7]. First described in 1995 by Nies and Silver in archaea, bacteria, and eukaryotes, these proteins—cation diffusion facilitator (CDF) transporters—are part of a large cation efflux protein family responsible for the mobilization of divalent metal cations such as Zn^2+^, Cd^2+^, Co^2+^, Fe^2+^, Ni^2+^, Mn^2+^, and possibly Cu^2+^ and Pb^2+^ [8,9,10]. In 2007, Montanini et al. proposed the subdivision of cation efflux protein family members into three main groups based on their specificity for the transported metal: Group 1, Mn-CDF; Group 2, Fe/Zn CDF with the Fe^2+^ and Zn^2+^ substrates and other metal ions; and Group 3, Zn-CDF with Zn^2+^ substrates and other metal ions, but not including Fe^2+^ or Mn^2+^ [11]. Later, in 2013, Cubillas et al. refined the accuracy of this classification and suggested a new division, separating the CDF proteins into 18 independent clades based on the specificity of the transported metal [12]. Recently, Xu et al. identified the first CDF protein that, in addition to mobilizing Zn^2+^ ions, mobilizes Na^+^ ions, proposing the incorporation of a new CDF group: Na-CDF [13].

Transition metals such as zinc, iron, and copper are essential in all organisms. Their chemical characteristics permit them to be essential in different cellular processes such as metabolism, nutrient transport, free radical detoxification, or photosynthesis in chloroplasts [1,7,11,14,15,16]. A high environmental concentration of metallic ions, along with some other similar but non-essential metals such as lead, silver, mercury, or cadmium, can cause essential metals to become highly toxic due to competition reactions, which produce an increase in oxidative damage due to reactive oxygen species generation [15].

It seems that organisms conserve complex regulatory mechanisms that safely transport these ions, causing cells to reach tolerance to higher levels of ion concentrations. These mechanistic activities create a network of events that enable ionic absorption, extrusion, chelation, traffic, and storage, maintaining the metals’ homeostasis at the cellular level while also reducing potential cell damage [1,11].

Metal metabolism is carried out by specific transporters that are encoded in multigene families [8]. For 90 years, three leading gene families have been identified as involved in these processes: the P-type ATPase family that mediates the transport of Cd^+^, Cu^2+^, H^+^, K^+^, Na^+^, Mg^2+^, and Ca^2+^ ions at the expense of ATP hydrolysis, as described for archaea, bacteria, and eukaryotes [17,18]; the ABC transporter family that transports Ni^+^, Mn^+^, Fe^+^, Mo^+^, peptides, and sugars in archaea, bacteria, and eukaryotes [19,20]; and the RND transporter family that mobilizes Ni^2+^, Co^2+^, Cd^2+^, and Zn^2+^ ions and has only been reported in bacteria [18,21,22]. However, several heavy metal transporters that do not belong to the previously established groups have been reported in recent years. One of these is the cation diffusion facilitator (CDF) transporter family. The reported substrates of its members are divalent metal cations such as Zn^2+^, Co^2+^, Fe^2+^, Cd^2+^, Ni^2+^, and Mn^2+^, or those with an atomic radius ranging from 72 pm (Zn^2+^) to 97 pm (Cd^2+^) [6]. Most of the reported CDFs work with a two-flux metal+/H+(K+) transport system that catalyzes the efflux of transition metal cations from the cytoplasm to the outside of the cell or subcellular compartments, resulting in their increased tolerance to the presence of these cations [8,23,24,25,26].

The CDF transporter protein structure comprises 300 to 400 aminoacidic residues structuring six transmembrane domains (TMD) called TMD I-VI. These proteins have carboxyl and amino-terminal ends oriented towards the cytoplasm [9,11]. Paulsen and Saier proposed a characteristic sequence between the TMDI and TMDII domains and a C-terminal cation egress domain as members of the CDF family [10]. There is high conservation within the protein sequences corresponding to amphipathic TMDs I, II, V, and VI since they are possibly responsible for metal transport [27]. However, some reported members of the CDF family have exhibited significant differences on the level of their secondary structure; some examples are the Msc2 protein of *Saccharomyces cerevisiae* (*S. cerevisiae*), which has 12 TMDs, or the Znt5 protein of humans, which shows 15 TMDs [28,29]. Generally, the transporters of this family contain a histidine-rich region located between TMDs IV–V and at the N and/or C-termini. These regions are believed to have the ability to bind metals (Cd^2+^, Zn^2+^, and Co^2+^, among others), although they can be located in other areas; for example, the Zrg17 protein from *S. cerevisiae* is located in the ER, and this region is contained between TMD III and TMD IV, located towards the light of the ER. In contrast, Shcdf8 of *Stylosanthes hamata* does not contain domains rich in histidine; such variation is directly related to ion transport affinity [30,31]. CDF proteins are located in different membrane types, such as the cell membrane, and some are expressed in membranes belonging to specific compartments, such as the vacuole, Golgi, and endoplasmic reticulum [10,27]. In prokaryotes, they are usually associated with plasma membranes. In plants, the zinc transporters AtMTP1 and PtdMTP1 are located in the tonoplast or plasma membrane [26,32,33,34]. In mammals, the characterized proteins are unique to cytoplasmic vesicles [35,36,37].

Functional diversity also occurs in yeast, while in *S. cerevisiae*, the proteins encoded by the *ZRC1* and *COT1* genes have been reported in vacuoles [38]. In *Schizosaccharomyces pombe* (*S. pombe*), the *ZHF1* gene product has been found to be located in the endoplasmic reticulum [39].

*Yarrowia lipolytica* is one of the most significant yeast model organisms. It has genotypic and phenotypic differences from the other well-studied yeasts. Its genome contains 1229 unique genes, 1083 genes shared with *S. cerevisiae*, and 146 genes shared with other species. Functional analysis of the unique *Yarrowia* genes has revealed new strategies for molecule transport, ion homeostasis, and metabolic pathways [40]. Recently, bioremediation studies using *Y. lipolytica* have shown that this species can remove heavy metals such as nickel and chromium in the context of contamination [41,42], suggesting that these heavy metals’ remotion occurs through a chemical ion exchange mechanism involved in the biosorption process.

In this work, we present the structural and functional characterization of the *Yl*-Dmct protein encoded by YALI0F19734g of *Y. lipolytica* as an ion transporter through bioinformatic analysis with predictive methods in order to determine its possible origin and subcellular location by homology. We interpret its three-dimensional structure through protein modeling and describe and compare the physiological absence effect, including growth and morphology, between the Δ*dmct* mutant strain of *Y. lipolytica*, generated by deletion of the gene, the retromutant strain (R*dmct)*, constructed by insertion of the *DMCT* gene, and the parental strain under different culture conditions.

## 2. Materials and Methods

### 2.1. In Silico Analysis

Different web tools were used for the in silico characterization of the putative *Yl*-Dmct protein (YALI0F19734p) encoded by the *Y. lipolytica DMCT* gene. The primary search for structural homologues was carried out with Protein BLAST and the Conserved Domain Database of the NCBI (www.ncbi.nlm.nih.gov/cdd/, accessed on 6 January 2022) programs [43,44]. The functional analogous protein sequences were obtained using the ChimeraX software with the Protein BLAST tool (https://blast.ncbi.nlm.nih.gov/Blast.cgi?PAGE=Proteins, accessed on 6 January 2022) and the Uniprot (https://www.uniprot.org/, accessed on 7 January 2022) database, and they were compared with the Clustal Omega program (https://www.ebi.ac.uk/Tools/msa/clustalo/, accessed on 27 July 2022) [45,46,47]. The subcellular location was predicted with the DeepLoc 2.0 software (https://services.healthtech.dtu.dk/sevice.-php?DeepLoc-2.0, accessed on 2 August 2022) [48]. The transmembrane protein and localization were obtained with CCTOP (http://cctop.ttk.hu/job/submit, accessed on 10 August 2022) [49]. The prediction of the three-dimensional structure of *Yl*-Dmct was performed using the ChimeraX software (https://www.cgl.ucsf.edu/chimerax/download.html, accessed on 26 August 2022), equiped with the AlphaFold and ColabFold tools [50,51]. Confidence levels were determined according to pLDDT metrics. The Predicted Align Error (PAE) was determined with an Error Plot. To establish the phylogenetic origin of the *DMCT* (YALI0F19734g) gene and its possible orthologs, we used PhylomeDB (www.phylomedb.org/, accessed on 28 August 2022) [52]. The analysis of the *Yl*-Dmct protein regulatory region was performed using Yeastract (www.yeastract.com/, accessed on 4 September 2022) [53]. Genome mining of the genes possibly involved in the transport of metals was carried out using information from the *Y. lipolytica* CLIB122 genome (www.ncbi.nlm.nih.gov/genome/194?genome_assembly_id=28431, accessed on 28 September 2022) [54], and those genes with functional domains previously reported in the Conserved Domain Database (CDD) that are related to the transport of divalent cations were selected.

### 2.2. Microorganisms and Growth Conditions

The *Y. lipolytica* strains used in this work were P01a (MatA, Ura3-, Leu2-), supplied by C. Gaillardin, and Δ*dmct* and R*dmct* (MatA, Ura3-, Leu2-), generated in this work.

All strains were cultured in a solid or liquid YPD medium (1% yeast extract (Beckton Dickinson Bacto, Mexico City, Mexico); 2% peptone (MCD Lab, Mexico City, Mexico); 2% glucose (CTR Scientific, Monterrey, NL, Mexico); 2% agar (Beckton Dickinson Bacto, Mexico City, Mexico) when required); or YNB medium (Beckton Dickinson Bacto, Mexico City, Mexico) (0.67% YNB without amino acids (Beckton Dickinson Bacto, Mexico City, Mexico) supplemented with leucine (Jalmek, San Nicolás de los Garza, NL, Mexico) (262 mg/L) with or without uracil (Sigma-Aldrich, St. Louis, MO, USA) (22.4 mg/L), 0.5% glucose (Jalmek, San Nicolás de los Garza, NL, Mexico), and 2% agar, when required). The cultures were kept at 28 °C with shaking (200 rpm) overnight.

### 2.3. Construction of Δdmct Strain

Genomic DNA was isolated from the *Y. lipolytica* cells (P01a strain) after overnight culture using the method described by Hoffman and Winston [55]. The gene encoding the putative protein *Yl*-Dmct and its 5’ and 3’ ends were amplified with specific primers, TMn-F 5’-CTCGAGGTGATTACGCGGGGTGG-3’ and TMn-R 5’-CTCGAGCATTTAGGAAGCCTGGCC-3’, designed on the adjacent sequences 5’ and 3’ of the contig YALI0F19734g from *Y. lipolytica* (https://www.ncbi.nlm.nih.gov/gene/2908296, accessed on 15 may 2017). The obtained PCR product (2980 bp) was purified and cloned into a TOPO4 vector (TOPO4 TA Cloning Kit, Invitrogen, MA, USA).

The internal coding part of the YALI0_F19734 gene (1668 bp) was removed by digestion with *Xba*I and *Kpn*I restriction enzymes and replaced with the *URA3* gene, which contains their 5’ and 3’ flanking regions (1750 bp). The obtained product was used for the transformation of *E. coli* TOP10 cells. Mini-preps of plasmidic DNA from *E. coli* colonies grown in selective medium, LB with ampicillin, were obtained, and the selection of clones harboring the disruption cassette was performed by restriction analysis and corroborated by PCR with the specific primers mentioned above. Moreover, genetic construct insertion at the correct locus and in the correct orientation was PCR-validated with specific oligonucleotides, TMn5’-F 5’-GGCAAGTGCAATAGAGTTGGGTGTG-3’, designed at 300 bp of the chromosomal 5’ region, as well as the reverse oligonucleotide URA3-R 5’-CCTCGGCACCAGCTCGCAGGCC-3’ and the URA3-F forward oligonucleotide 5’-GGCCTG-CGAGCTGGTGCCGAGG-3’, both located in the ORF of *URA3* and TMn-R (described above) reverse oligonucleotide. The disruption cassette (2525 bp) was obtained by PCR with high-fidelity DNA polymerase (Invitrogen) and used for the transformation of the *Y. lipolytica* cells by the lithium acetate method [56]. The *Y. lipolytica* mutant cells’ selection was based on uracil prototrophy on YNB plates lacking uracil and confirmed by PRC reactions with specific primers.

### 2.4. Reintegration of the DMCT Gene

In order to obtain the complementary strain, DNA from the P01a strain was used to amplify the *DMCT* gene and its adjacent regions by PCR. For amplification, the oligonucleotides Tmn5’ (5’-CGCTCTATCCCCTACCCTAACCCG-3’) and Tmn3’ (5’-GGAGTTACCCTACGATCT CTCACGAG-3’), which were specifically designed, were used. The PCR product (3252 bp) was purified and used for the transformation of lithium-competent cells (Δ*dmct* strain) by the lithium acetate method coupled with heat shock [56]. Positive clones were selected by culture in a selective medium: YNB supplemented with leucine, uracil, and/or 5’ FOA (GoldBio Inc., St. Louis, MO, USA). Molecular confirmation of the positive clones was achieved by PCR with specific primers, and enzymatic digestion of the amplicon was carried out with *EcoR*I, *Xho*I, and *Sac*I restriction enzymes.

### 2.5. Phenotypic Characterization

Phenotype changes in the two strains during their culture growth in formulated media with different divalent cation concentrations were registered. To determine the divalent metal cations’ maximum concentration tolerance, cell dilutions (10^−^^1^ to 10^−^^5^) were inoculated in Petri dishes containing YNB medium supplemented with different Ca^2+^ concentrations (800–900 mM), Cu^2+^ (2–4 mM), Fe^2+^ (6–8 mM), Mg^2+^ (20–300 mM), Mn^2+^ (2–30 mM), or Zn^2+^ (16–20 mM). Base YNB medium without additions was used as a control.

For cell growth analysis and to assess physiological behavior at different cation concentrations, the cells (from all strains) previously grown overnight were inoculated (OD600 nm = 0.2 units) into flasks containing 50 mL of YNB medium amended with each of the following cations at variable concentrations: Ca^2+^ (650–800 mM), Cu^2+^ (2–8 mM), Fe^2+^ (2–8 mM), and Zn^2+^ (12–18 mM). As a control, cultures in YNB medium were used. Cell growth was quantified by spectrophotometry (OD600 nm), and the macroscopic and microscopic morphology of the cells was observed and documented with an OmaxMR phase-contrast microscope (OMAX Corp., Kent, WA, USA).

For the phenotypic characterization, in all the experimental series, divalent cations were used as chloride salts (or iron sulfate, when indicated), which were obtained from Jalmek, México.

### 2.6. Intracellular Accumulation of Cations

Patterns of intracellular cation accumulation were analyzed using fluorescence assays. Cells in the logarithmic growth phase were washed, resuspended in PBS buffer, and processed as follows: For divalent cation staining, 20 μL of cells were mixed with 5 μL of Phen green SK (Cayman Chemicals, Ann Arbor, MI, USA) working solution (50 μM). The cells were incubated for 1 h in the dark, washed twice with PBS, resuspended in 20 µL PBS, and observed under a Leica DMRE fluorescence microscope (Leica, Wetzlar, Germany) adjusted to a 507/532 nm absorption/emission range. Cell wall staining analysis was performed using the cells previously washed with PBS (10 μL), which were then mixed with 1 μL of Calcofluor white (Sigma-Aldrich, Darmstadt, Germany) working solution (Calcofluor white: KOH 10%: PBS 1X 1:1:3). The samples were observed under the fluorescence microscope at a 360/430 nm wavelength. The images were processed using the Leica Application Suite (LAS) V4.0 software.

### 2.7. Statistical Analysis

All experiments were performed in triplicate, and the data are expressed as the mean +/− standard error. Statistical analysis was performed using one-way ANOVA with the package SPSS v20.3.

## 3. Results

### 3.1. In Silico Yl-Dmct Protein Analysis

According to our comparative genomics analysis, in the genome of *Y. lipolytica*, there is only one copy of the *DMCT* gene, and no homologues were found. The *DMCT*-encoding gene (YALI0F_19734g) from *Y. lipolytica* has an Open Reading Frame (ORF) of 1668 nt. This gene encodes a predicted protein of 556 amino acid residues. Multiple alignment analysis of the putative protein encoded by the *DMCT* gene permitted us to identify regions that present similarities between this protein and possible homologous proteins in other species. The results in Table 1 show that, except for the CDF family transporter identified in *Candida albicans* (*C. albicans*), the rest of the proteins included in the comparative analysis showed similarity percentages ranging from 19.01% to 23.2% among the complete proteins and 18.67% to 24.24% in the restricted CDF domain regions. We observed a higher similarity percentage between the *Yl*-Dmct protein and the protein corresponding to the possible homologous cation diffusion facilitator protein of *C. albicans* (44.3% and 56.77%, respectively).

The multiple alignments of the putative protein encoded by the *DMCT* gene of *Y. lipolytica* allowed us to identify regions with similarities between this protein and possible homologous-orthologous proteins in other species. The results are shown in Figure 1, where the 38 conserved amino acids are shown and asparagine, glutamine, glycine, and serine are highlighted. The asparagine quartet at positions 301, 305, 410, and 414 of *Yl*-Dmct is conserved in all possible homologues, even when the percentages of similarity are low. We observed the highest similarity percentage of the *Yl*-Dmct protein with the hypothetical protein from *Galactomyces reessii*, which contains the FieF domain, characteristic of transporters of divalent metal cations (Fe/Co/Zn/Cd).

Taking into account the low percentages of similarity between the *Yl*-Dmct protein and its structural homologues, we decided to compare it with proteins that have a similar function, classifying them as functional homologous-analogous, taking into account the fact that the proteins were described as diffusers of iron, zinc, and copper cations and that the selected organisms were evolutionarily close to *Y. lipolytica*, taking amino acid sequences from *S. cerevisiae*, *Candida glabrata*, and *C. albicans*. Finally, taking into account what was discovered later in subcellular localization, the search discriminated against only those transporters located in vacuoles or lysosomes and in the Golgi apparatus. In the results shown in Table 2, we found that despite the functional analogy and cellular co-localization, the transporters share very little similarity with the *Yl*-Dmct protein, with Cot1 and Ctr2 from *S. cerevisiae* being the most and least similar, with 16% and 9% identity, respectively.

### 3.2. In Silico Yl-Dmct Analysis: Cellular Location, Phylogenetic Analysis, and Three-Dimensional Modeling

*Y. lipolytica* Dmct protein encoded by the *DMCT* gene was determined by employing a numerical matrix. Using a matrix, the problem protein is compared with proteins registered in databases by the neighbor-joining method to predict those with which it shares a location, based mainly on the protein sequences and the functional motifs. In this way, in Figure 2A, we can observe that according to the resulting values, the most likely location of the *Yl*-Dmct protein is the lysosome/vacuole, with a value of 0.8105 compared to the threshold of 0.5848. Localization in the Golgi apparatus is also likely. However, the difference between the probability and the threshold is low, being 0.6821 and 0.6494, respectively.

Analysis of the putative *Yl*-Dmct protein showed that it consists of 555 amino acids; of these, 115.91 amino acid residues are contained within the membrane, forming five strong transmembrane domains (Figure 2B in yellow) and one weak transmembrane, which cannot cross from one side of the membrane to the other (Figure 2B in orange). The results suggested an 89% probability and that the N-terminal end is oriented towards the cytoplasmic region.

Phylogenetic analysis showed that the *Yl*-Dmct protein is closely related to other yeast transporters, highlighting the genera *Geotrichum*, *Clavispora*, *Debaryomyces*, *Pichia*, and *Ogataea*. In the phylogenetic approach to the possible origin of the *DMCT* gene (Figure 2C), it is interesting to observe the three duplication events and three speciation events that indicate *U. maydis* as the origin and common ancestor. The presence of this gene and its predecessor variants is also highlighted in both basidiomycetes and ascomycetes.

In the three-dimensional model, the conformation of the *Yl*-Dmct protein can be observed, and a classic transporter structure is demonstrated. It contains six transmembrane domains represented by lower alpha helices (Figure 3A), which, in turn, are made up of amino acids that give it a hydrophobic nature (Figure 3C), typical of a transmembrane waistband. On the other hand, we highlight that around 65% of the three-dimensional predictions have a reliability between 90% and 100%.

Metals generally use trigonal or quartet coordination in their transmembrane domains. The transporter site between the six transmembrane domains (Figure 3B) enables cation diffusion due to a quartet of asparagines at sites 301, 305, 410, and 414, which are located in a mirror-like manner in TM2 and TM5 (Figure 3D), which preserves the metal binding site in the transmembrane region (Figure 3E).

### 3.3. In Silico Analysis of Putative Binding Sites for Transcription Factors in the Promoter Region from the Yl-DMCT Gene

Sequence analysis, corresponding to the putative promoter region 1000 nt before the ORF start site of the *Y. lipolytica DMCT* gene, allowed us to identify 97 possible binding sites for 42 different transcription factors (Table 3). Among the predicted binding sites are those corresponding to transcription factors (TF) related to various cellular processes, such as the response to oxidative stress (Yap1, Skn7p, Msn2, and Msn4), the response to high metal concentrations (Aft2p, Cup2p, and Stb4p), and the pH response (Nrg1).

### 3.4. Δdmct and Rdmct Strains’ Phenotypic Characterization

The *Yl*-Dmct protein’s association with cation transport in *Y. lipolytica* was approached using constructed mutant cells generated by deletion of the *DMCT* gene, as described in the Materials and Methods section.

A qualitative analysis of the growth of cells corresponding to the P01a (wild type), Δ*dmct* (mutant), and R*dmct* (retromutant) strains was performed. The *Y. lipolytica* cells in the exponential growth phase were cultured on YNB plates containing different concentrations of divalent cations (Mg^2+^, Mn^2+^, Cu^2+^, Fe^2+^, Zn^2+^, and Ca^2+^). In these experiments, three R*dmct* clones (016, 017, and 021) were included and compared with mutant and wild-type strains (Figure 4).

The obtained results showed that of all the tested cations, magnesium and manganese did not show any significant difference between strains, while high calcium concentrations had a minor visible effect on the macroscopic growth rate of all the strains. This effect was not dependent on the calcium concentration because slight growth inhibition was observed at all the tested concentrations (800–900 mM). Growth inhibition was similar across all the strains. These results suggest that the *Yl*-Dmct protein is not essential for calcium transport.

On the other hand, when other cations (Cu^2+^, Fe^2+^, and Zn^2+^) were added to the culture medium, the cell growth of all the strains was inhibited. This effect was more significant in the presence of Cu^2+^ (2–4 mM), whereas at different concentrations of Fe^2+^ (6–8 mM) and Zn^2+^ (16–20 mM), growth inhibition was directly proportional to the concentration of the cations, and higher inhibition was observed at the higher concentrations of Fe^2+^ (6.5–8.0 mM) and Zn^2+^ (20 mM). In all cases, the mutant strain (Δ*dmct*) showed minor growth with respect to the parental strain in the presence of Cu^2+^, Fe^2+^, or Zn^2+^.

Cell growth quantitative analysis of the parental (P01a), mutant (Δ*dmct*), and retromutant (R*dmct*) strains was carried out by culturing cells in YNB liquid medium supplemented with different calcium, copper, iron, and zinc concentrations. The growth rate was determined by spectrophotometry (OD600 nm) after 20 h of culture. Figure 5 shows the results of this analysis. In the presence of calcium or copper, the three strains’ growth showed the same tendency to decrease as the concentration of calcium (Figure 5A) or copper (Figure 5B) in the culture medium increased. However, it is interesting to note that in the presence of copper (4 mM), the Δ*dmct* strain’s growth was four times higher than that of the P01a and R*dmct* strains. In the presence of iron, yeast inoculum precipitated the iron (iron sulfate) present in the medium; thus, the spectrophotometric determination was compromised (Figure 5C). When the culture medium was zinc-amended, the P01a and R*dmct* strains’ growth was similar to that of the calcium- or copper-containing cultures, showing the same tendency to decrease growth with the increased zinc concentration (Figure 5D). In contrast, the growth of the mutant strain (Δ*dmct*) was constant. It did not decrease with the increased zinc concentration (12–18 mM), suggesting that the *Yl*-Dmct protein plays an essential role in sensing and/or the tolerance response to elevated zinc concentrations.

### 3.5. Cell and Colony Morphological Characteristics

Cells growing (exponential phase) in media supplemented with different ion concentrations were observed by microscopy, and differences in growth were noted. In the YNB medium without cations, both strains grew in yeast form. In the presence of calcium (650 and 700 mM), cells corresponding to the mutant strain (Δ*dmct*) presented morphological changes (growth as mycelium); these observations suggested that calcium induced dimorphic changes (mycelium formation) in the mutant strain. However, these morphological changes were repressed in the cells growing at higher ion concentrations (750–800 mM). In the presence of copper or iron (Figure 6), the cultures of both strains (P01a and Δ*dmct*) showed cell agglomeration that did not permit a complete visualization of the cell morphology. In the presence of iron (2–8 mM concentration), when cells corresponding to both strains were inoculated, it was observed that soluble iron (added as iron sulfate) formed a precipitate after 20 h of incubation of the cultures.

When zinc (12–18 mM) was added to the culture media, it was observed that low zinc concentrations (12–14 mM) do not induce changes in cell morphology, and cells of both strains remained as yeast, whereas at higher zinc concentrations (16–18 mM), the cells corresponding to the Δ*dmct* strain but not the P01a strain presented morphological changes that induce vegetative growth (cells forming a mycelium).

On the other hand, when the colonial morphologies corresponding to the parental (P01a) and mutant (Δ*dmct*) strains were examined, differences in both the size and form of the colonies were noted when the cells were cultured in the presence or absence of different ions (Figure 6). In the control medium (YNB), the P01a colonies presented a creamy white consistency and a cerebriform surface with an entire/filamentous margin. In contrast, colonies from the Δ*dmct* strain showed a creamy, white consistency, a crater-shaped surface, and a filamentous margin.

In the presence of calcium (80 mM), similar development patterns were observed. The colony surfaces were completely filamentous. Interestingly, as the calcium concentration increased, the branches’ length decreased (data not shown). When cooper (2–8 mM) was added to the culture medium, the colonies from both strains could form long and disordered filaments, which were notably longer in the colonies from the mutant strain at the 4 mM concentration.

On the plates containing iron, slight differences in the development of the strains were observed. At low concentrations (2–4 mM), colonies corresponding to the P01a strain showed filamentous/granular growth, while with 6 mM iron, the viability of the cells was significantly affected. Additionally, in colonies corresponding to the Δ*dmct* strain, it was observed that low iron concentrations produced colonies with a filamentous/granular growth pattern, which was similar to that observed in the P01a strain. When the iron concentration reached 6 mM on the plates, the cell viability of colonies from the mutant strain was compromised, suggesting a greater susceptibility of this strain to the presence of iron.

Zinc addition led to the growth of colonies from both strains in the form of a long and disorderly-branched mycelium. As the Zn^2+^ concentration increased on the plates, it was observed that the colonies formed were similar to those of the P01a strain. The colonies from both strains presented filaments at the edges, with these structures being more prominent in colonies corresponding to the Δ*dmct* strain.

### 3.6. Intracellular Accumulation of Divalent Metal Cations and Cell Wall Integrity

To evaluate changes in the intracellular metal accumulation patterns in both the presence and absence of the *Yl*-Dmct protein, the P01a and Δ*dmct* strains were cultured in a YNB medium containing different cation concentrations. Furthermore, fluorescence microscopy assays were performed using the following fluorophores: Phen green SK diacetate with affinity for intracellular divalent metal cations or Calcofluor white with affinity for cell wall polysaccharides.

Figure 7 shows the results obtained for the unstained cells (bright-field microscopy) and cells stained (fluorescence microscopy) with the different fluorophores (as indicated). The images in Figure 7A correspond to cells from the P01a and Δ*dmct* strains cultured in YNB without cations. Under all the staining conditions, one can observe a mixture of cells that present a yeast morphology or a slightly elongated form. Similar growth patterns were observed in the parental (P01a) and mutant (Δ*dmct*) strains when the cells were cultured in the presence of 16 mM zinc (Figure 7B). In the first panel, it can be observed that the cells of both strains presented a similar morphology: yeast-like or slightly elongated cells, with some forming a septate mycelium. Staining with Phen green SK permitted us to observe that the intracellular distribution of zinc was homogeneous, mainly in the cytoplasm. However, accumulations in vacuoles were distinguished in some cells from both the parental and mutant strains. Calcofluor white staining showed that although some cells (from both strains) were not entirely differentiated, they maintained cell wall integrity.

In the presence of 4 mM copper, *Y. lipolytica* cells from the parental strain showed a slightly elongated form. This morphology was visible in cells stained with Phen green SK and Calcofluor white fluorophores. In contrast, in the cultures corresponding to the mutant strains, the presence of large cell aggregates was evident (Figure 7C, panel 1). The analysis of cells stained with Phen green SK fluorophore (Figure 7C, panel 2) showed that copper incorporated into the cells from the parental strain was homogeneously distributed; in contrast, in the mutant cells (Δ*dmct*), there were areas of accumulation of fluorescence, suggesting copper accumulation in cells lacking the *DMCT* gene. Staining with Calcofluor white (Figure 7C, panel 3) did not show differences in the morphology of the strains, demonstrating that deletion of the *DMCT* gene does not induce structural changes in the cell wall.

As expected, the cells growing in culture media without metal cations were not stained with Phen green SK, while in the presence of the different divalent cation cells, both the parental and mutant strains were stained with Phen green SK; however, the obtained results suggested no differences in intracellular cation accumulation between the strains. Interestingly, in the cells cultured in the presence of iron (4 mM), the accumulation of this cation showed a more homogeneous distribution (Figure 7D) than the cation distribution in cells cultured in media containing other cations.

### 3.7. Identification of Putative Genes That Participate in the Transport of Metals as a Possible Compensatory Response to Dmct Deletion

With the experimental approaches used, a clear phenotype that indicates the function of the protein encoded by the *DMCT* gene was not observed in the mutant strain (Δ*dmct*). These results suggest that the absence of the *Yl*-Dmct protein triggers the possible compensatory activity of other proteins involved in metal transport pathways. Under these considerations, we decided to analyze, *in silico*, the presence of other genes with functional domains related to the canonical and alternative transport of metals present in the *Y. lipolytica* genome that could be involved in the transport and trafficking of metals. As a result, 27 genes scattered across the six chromosomes were identified (Table 4). None of these putative genes or the activity of the encoded proteins have been described thus far.

## 4. Discussion

*Yarrowia lipolytica* is a yeast of great interest for the biotechnology industry. Moreover, in recent years, it has been used as a model organism to study different biological processes. Concerning the transportation of metals, it is still unknown how these species realize the transport of cations to maintain cell homeostasis. In this work, we analyzed the structural characteristics of the *DMCT* gene (YALI0F19734g) and the putative encoded protein, as well as the effects of the absence of this protein (*Yl*-Dmct) on growth, dimorphism, and specificity in ion transport.

First, the bioinformatic analysis revealed that there are extremely low percentages of conservation between structurally homologous proteins (19–44% in the total protein and 19–56% in the CDF domain) and functional analogues (9–16%) (Table 1 and Table 2). Haney et al. [27] reported that the degree of sequence conservation of member proteins of the CDF family is extremely low, while Liesegang et al. [64] reported similarity percentages ranging from 28% to 46% in some cases. The multiple amino acid alignment (Figure 1) revealed the conservation of 38 amino acids, among which asparagine, glutamine, glycine, and serine stand out. An asparagine quartet is conserved in all the possible structural homologues. Barber-Zucker et al. [65] mentioned that Ca^2+^ ions are the most abundant metal bound to the quartet of asparagines (Figure 3). Studies of the ZNT4 gene in mammals have shown that CDF transporters are also associated with homo-oligomeric complexes or hetero-oligomeric complexes, interacting directly with other proteins that require metal cations for their operation [65,66,67,68]. Nevertheless, as reported by Lu et al. [69], modeling of the diffusing protein of cations through tempering generates a homodimer that uses Zn^2+^ as an attachment point anchored to the residues His519, Glu523 and Asp543; however, in the three-dimensional *Yl*-Dmct model, the amino acids that function as a Zn^2+^ binding site are not co-located close to the dimeric formation site because His519 and Glu523 are located in an alpha helix and Asp543 in a beta strip. Therefore, there is no opportunity for Zn^2+^ to bind and form a homodimer.

Secondly, the *Yl*-Dmct protein from *Y. lipolytica* presents several transmembrane regions, which suggest that it may bind both extra- and intracellular membranes such as the lysosomes, vacuoles, and the Golgi apparatus (Figure 2A,B), as observed in prokaryotic cells [27] or in other yeast cells, such as *S. pombe* [39]. The predicted conformational tertiary structure of the *Yl*-Dmct protein (see Figure 3) is similar to the proteins identified as possible homologues, such as Cot1, Zrc1, and Zrg17, which have been described as involved in the vacuolar transport of zinc and other ions [70].

The *Y. lipolytica* Dmct protein’s phylogenetic analysis showed a close relationship with genes present in other yeast genera such as *Geotrichum*, *Clavispora*, *Debaryomyces*, *Pichia*, and *Ogataea*, among others. Most of these are not fully characterized organisms; however, a particular characteristic is shared by these five genera: tolerance to various environmental scenarios that are stressful or even lethal for any other organism. For example, the genus *Debaryomyces* is characterized as osmotolerant, halotolerant, and xerotolerant [71], and it has been described that the tolerance of these organisms is due to the secretion systems present with them, which are constituted by transmembrane transporters belonging to the CDF family. These transporters are responsible for mobilization to the outside of the cell or internal storage of different stress agents to avoid their accumulation, which could generate toxicity conditions [72]. In several studies, *Y. lipolytica* has been considered a polyextremophilic yeast that is tolerant to stress conditions generated mainly by the environment [73,74]. Hence, the presence of the putative protein encoded by the *DMCT* gene in *Y. lipolytica* allows us to consider that this protein is involved in transport and/or tolerance to metals, a fact that has been confirmed for other gene members of the CDF family that have been described in other organisms [8,23].

The sequences adjacent to the *Yl*-Dmct protein contain 97 possible binding sites for 42 different transcription factors (TF), of which the most abundant are those related to the stress response. The presence of such TFs could be explained by the fact that, as a response to different stress conditions that compromise essential cellular functions, *Y. lipolytica* cells activate the transcription of the *DMCT* gene to counter the effect(s) of a possible excess or lack of cations. It is essential to highlight the presence of specific TFs: Stb4, which is involved in the transcription of transmembrane transporters in general; Aft2, which, in the absence of iron, is responsible for the transcription of genes whose products mobilize copper [75]; Yap1, related to cadmium tolerance, which is involved in the transcription of genes that respond to ionic detoxification, in addition to the expression of ferredoxin and ferredoxin reductase proteins [76]; and Cup2, which is similar to Yap1 and activates the transcription of metallothionein proteins when the cell is in an environment that presents excess metals. These proteins “trap” the ions and keep them occupied, thus avoiding oxidation-reduction reactions due to free ions [77]. Other possible sites were found to correspond to the binding of TF, such as Nrg1, and these are related to the response to stress due to pH changes. pH is of great importance in the chemistry of metals because its modification promotes metals such as iron, copper, and cadmium, among others, to change their oxidation state, which results in the generation of free radicals and oxidative damage to the cell [78]. Thus, the activation of other TFs such as Yap1, Skn7, Msn2, or Msn4 regulates genes involved in the antioxidant cellular response. The presence of these sites that, presumably, are implicated in *Yl*-Dmct protein regulation suggests that the product of this gene is implicated in responses to different factors causing stress for *Yarrowia* cells.

To analyze the function of the protein encoded by the *DMCT* gene in *Y. lipolytica*, a mutant strain was generated by deletion of the *DMCT* gene. Additionally, the complementary strain (R*dmct*) was generated by the reintegration of the *DMCT* native gene.

In the qualitative analysis, all *Y. lipolytica* strains showed high tolerance to calcium, growing in concentrations of up to 880 mM of calcium chloride. It could be observed that cell growth showed a direct relationship to the calcium concentration. In 1997, Beeler et al. reported that *S. cerevisiae* could grow and develop normally at concentrations as high as 100 mM CaCl_2_ [79]. Our results showed that the growth of *Y. lipolytica* in the presence of calcium was almost nine times higher, which demonstrates its greater capacity to tolerate osmotic stress.

In the presence of zinc (16–18 mM), cells corresponding to the mutant strain showed minor growth (with respect to the parental strain), but in the presence of copper (4 mM), the growth of the Δ*dmct* strain was four times higher than that of the P01a and R*dmct* strains. This phenomenon was not observed when high Cu^2+^ concentrations (6–8 mM) were used. In general, both the parental and mutant strains were susceptible to the highest concentrations tested but showed tolerance to low cation concentrations. Tolerance to these cations has been demonstrated in other yeast species, such as *Z. rouxii* and *S. cerevisiae*, which showed tolerance to concentrations in the range of 1 to 4 mM [80]. Comparing other phylogenetically related yeasts in terms of their tolerance to divalent cations, we can infer that *Y. lipolytica* has a greater tolerance to ionic stress. This greater tolerance may be due to the presence of adaptation mechanisms developed during the evolution and expansive duplication events of *Y. lipolytica* [81], assuming the presence of many genes that encode proteins that participate in cation transport mechanisms in its genome.

When *Y. lipolytica* cells were cultured in both solid and liquid media with the addition of calcium, both the parental and mutant strains presented morphological changes characterized by mycelium production. Ruiz-Herrera and Sentandreu [82] reported the effects of different dimorphism effectors on *Y. lipolytica*. The authors described that *Y. lipolytica* cells exhibited a phenotype similar to that observed in the presence of calcium and attributed it to changes in pH values, where the mycelial percentage increased by up to 90% in the pH range from 5 to 7. Based on these observations, we can assume that high calcium concentrations in the culture medium can affect intracellular pH, and therefore, changes in the differentiation patterns of *Y. lipolytica* cells are due to changes in intracellular pH and are not directly related to the effect of the calcium concentration in the culture medium or the absence of the *Yl*-Dmct protein. Taking the above into account, we suggest that possible changes in the pH values of the culture medium could be monitored or buffered for future determinations.

Interestingly, our results showed that *Y. lipolytica* cells from the mutant strain cultured in the presence of copper (2 mM) activated myceliation mechanisms. At the same time, at concentrations of 4–6 mM or higher, this process was repressed. This behavior suggests a copper-dependent induction-repression mechanism for dimorphism processes. In 2002, Silóniz et al. noted that high copper concentrations induce dimorphism in various yeasts [83]. In *Pichia gillermondi*, myceliation was observed at up to 0.65 mM Cu^2+^; when the copper concentration increased, an adaptive effect was observed due to the induction of accumulation mechanisms. In agreement with this, our results indicate that *Y. lipolytica* demonstrates dimorphic changes at a concentration of 2 mM Cu^2+^, its concentration limit before the bioaccumulation process of this ion begins, and morphogenesis mechanisms are suppressed at higher concentrations.

When iron was present in the culture medium, cells of both the P01a and mutant strains tended to precipitate the metal in the form of salts and form clusters with this compound. This phenomenon was reported by Safarik et al., referring to this behavior as cellular magnetic modification [84]. This mechanism is widely used to extract metal ions in various biotechnological processes [85]. The yeast genera that stand out for this characteristic are *Saccharomyces*, *Yarrowia*, *Klyuveromyces*, and *Rhodotorula*.

The morphological changes observed in the parental and mutant cells are associated with copper, iron, and intracellular zinc distribution. In cells from the parental strain, a homogeneous distribution of copper was observed throughout the cell structure. On the other hand, in the cell corresponding to the mutant strain, the copper distribution was not homogeneous, and ions could be observed forming clusters at very delimited sites. A similar phenomenon was observed when the cells were cultured in the presence of iron or zinc. In both cases, the presence of differentiated cells (mycelium) was observed, although such cells present with a lower intensity in their emitted fluorescence, which indicates a lower intracellular cation concentration. Such observations suggest that a decrease in the intensity of fluorescence in mycelial cells with respect to those that maintain the yeast form is due to a deficiency in the transport of cations.

Finally, although the collective results presented show that the protein encoded by the *DMCT* gene regulates the transport of divalent cationic metals specifically but is not generalized to non-metallic cations such as calcium, these findings also indicate that their activity is not specific to any metallic cation. In a more in-depth analysis, we observed that the *Yl*-Dmct protein does not have the histidine-rich regions reported by Gaither and Eide [86] for CDF proteins. The low transport specificity may be due to the absence of these amino acid residues, as mentioned by Haney et al. [27].

In general, we related the presence of the *Yl*-Dmct protein to conditions where the presence or absence of metal ions can threaten cell viability, considering the fact that, according to its characteristics, the protein encoded by the *DMCT* gene has the primary function of mitigating possible cell damage through the mobilization/transport of these ions. However, the presence of different TF-specific sites in the *DMCT* regulatory region suggests that this gene can respond when *Y. lipolytica* cells are exposed to different stress conditions. Additionally, this is not the only gene in charge of this critical task; bioinformatic analysis permitted us to identify 26 genes (Table 4) that are possibly also involved in the uptake, mobilization, and metabolism of metal ions in this species of great biotechnological importance.

## 5. Conclusions

The *Y. lipolytica DMCT* gene (YALI0F19734g) encodes a transmembrane protein involved in the transport of divalent cations. However, it does not demonstrate substrate specificity. The *Yl*-Dmct protein is involved in the homeostatic balance of copper, zinc, and iron but not calcium. Deletion of the *DMCT* gene caused cell morphological changes in response to extreme conditions induced by ionic treatments. Additionally, mutant cells could tolerate higher zinc and copper concentrations compared with the parental strain. However, *DMCT* is not a vital gene for the transport and homeostasis of zinc, copper, iron, and calcium ions. Other genes that may possibly complement the *Yl*-Dmct function are present in the *Y. lipolytica* genome. The results presented here suggest that, as mentioned repeatedly, the *DMCT* gene (YALI0F19734g) from *Y. lipolytica* encodes a protein whose characteristics permit it to be considered a member of the Cation Difussion Facilitador (CDF) family. Furthermore, this represents the first report about the characterization of a member of the CDF family in this species of great biotechnological importance.

## Figures and Tables

**Figure 1 jof-09-00600-f001:**
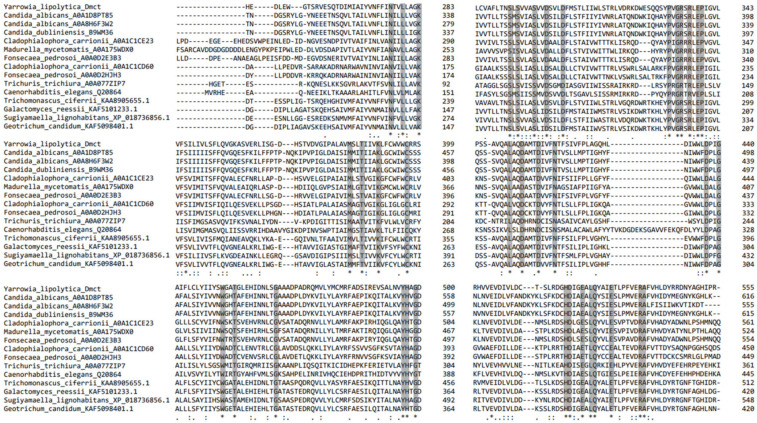
Multiple alignments of possible structural homologues of the putative protein encoded by the *Yarrowia lipolytica DMCT* gene. * Indicate conserved amino acids in all analyzed proteins.

**Figure 2 jof-09-00600-f002:**
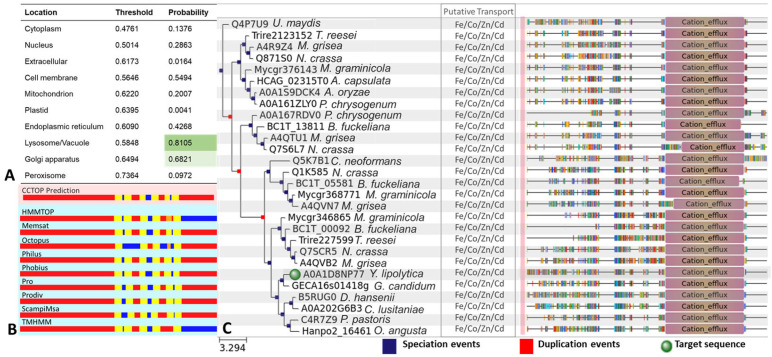
In silico analysis. (**A**) Prediction of the subcellular localization of the *Yl*Dmct protein. Probability values range from 0 to 1, where 0 is not at all likely and 1 is very likely, and those that are above the threshold are considered reliable. (**B**) Prediction of the conformation of transmembrane domains. Red lines represent internal/cytosolic regions; yellow encodes transmembrane regions; orange shows membrane reentry loops; and blue represents external/extracytosolic regions. (**C**) Philoma of the *DMCT* gene. The figure was generated with Phylome DB software. *U. maydis*, *Ustilago maydis*; *T. reesei*, *Trichoderma reesei*; *M. grisea*, *Magnaporthe grisea*; *N. crassa*, *Neurospora crassa*; *M. graminicola*, *Mycosphaerella graminicola*; *A. capsulate*, *Ajellomyces capsulata*; *A. oryzae*, *Aspergillus oryzae*; *P. chrysogenum*, *Penicillium chrysogenum*; *B. fuckeliana*, *Botrytis fuckeliana*; *C. neoformans*, *Cryptococcus neoformans*; *G. candidum*, *Geotrichum candidum*; *D. hansenii*, *Debaryomyces hansenii*; *C. lusitaniae*, *Clavispora lusitaniae*; *P. pastoris*, *Pichia pastoris*; *O. angusta*, *Ogataea angusta*.

**Figure 3 jof-09-00600-f003:**
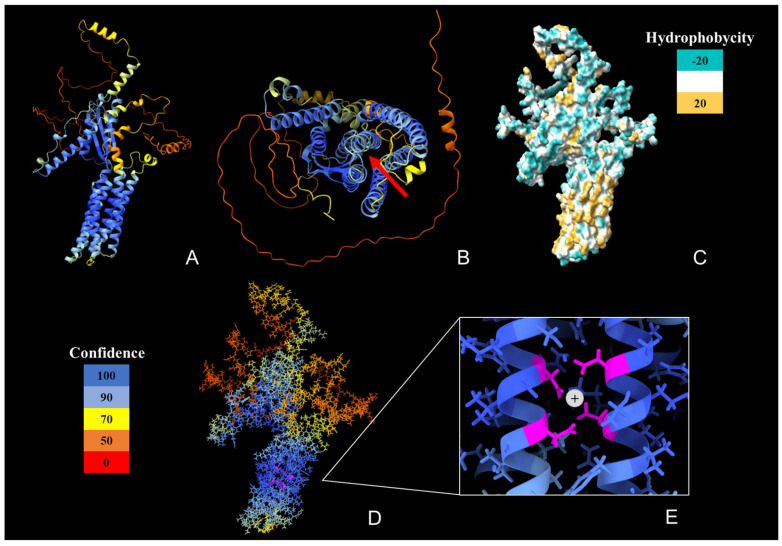
Three-dimensional modeling of the *Yl*Dmct protein performed with AlphaFold. Reliability ranges from 0 (red) to 100 (blue). (**A**) Longitudinal view; it shows the secondary structures, six alpha helices corresponding to the transmembrane domains, and folded beta strips. (**B**) Cross-sectional view; the red arrow points to the hole between the six transmembrane domains. (**C**) Hydrophobic model. The hydrophilic zones are blue, while the hydrophobic zones are gold. (**D**) Longitudinal view of the molecular model; the quartet of asparagine residues (ASP301, ASP305, ASP410, and ASP414) that allow the diffusion of cations are indicated in purple. (**E**) Close-up of the quartet area and representation of the interaction it has on cation diffusion.

**Figure 4 jof-09-00600-f004:**
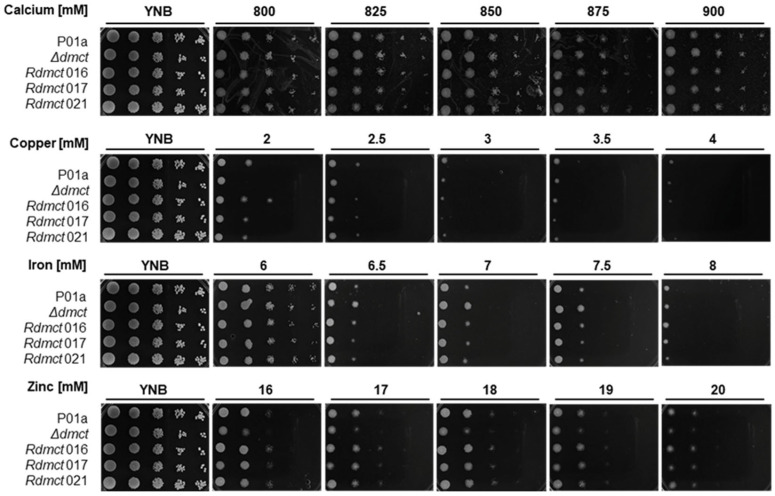
Growth of *Yarrowia lipolytica* P01a, Δ*dmct*, and R*dmct* strains in YNB-plates added with different concentrations of calcium, copper, iron, or zinc. The images correspond to cell growth in plates cultured for 72 h under the previously described conditions. Three retromutant strains (R*dmct* 016, 017, and 021) were considered for this experiment.

**Figure 5 jof-09-00600-f005:**
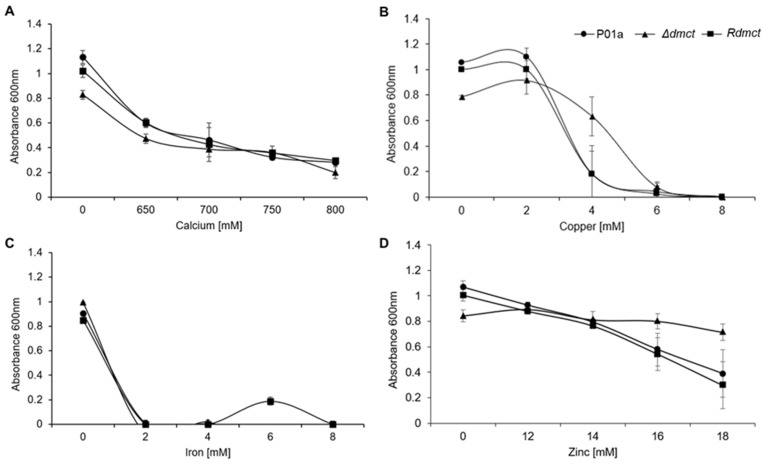
Growth of *Yarrowia lipolytica* P01a, Δ*dmct,* and R*dmct* strains, in YNB medium added with different (**A**) calcium. (**B**) Copper, (**C**) iron, (**D**) and zinc concentrations. Growth was determined by the cell density (OD600 nm) of cultures at 20 h. Plotted values correspond to the three experiments’ average; bars over each point correspond to the standard error.

**Figure 6 jof-09-00600-f006:**
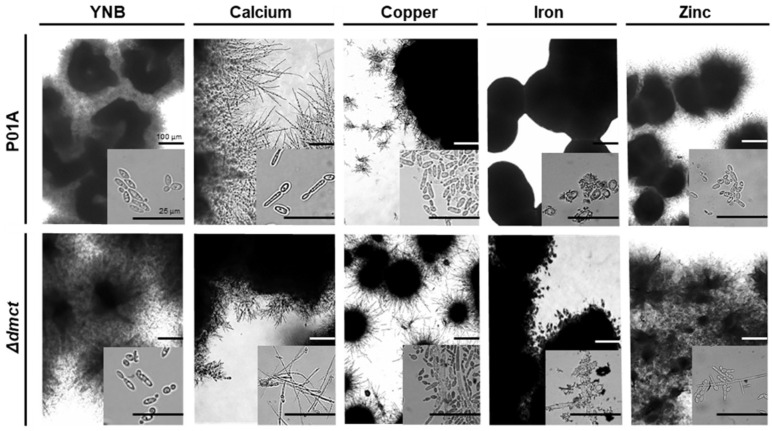
Morphology of *Yarrowia lipolytica* P01a and Δ*dmct* strains growing in YNB medium added with calcium, copper, iron, and zinc. The colonies (outer boxes) correspond to 36 h cultures in YNB-plates with the addition (or not) of calcium (100 mM), copper (4 mM), iron (2 mM), and zinc (6 mM). Images of the inner box correspond to cells cultured for 20 h in YNB-liquid medium added (or not) with calcium (700 mM), copper (4 mM), iron (6 mM), and zinc (16 mM). Images were taken with an Omax phase-contrast microscope under brightfield microscopy at 10× (outer boxes) or 100× (inner boxes) magnification. Representative images are shown.

**Figure 7 jof-09-00600-f007:**
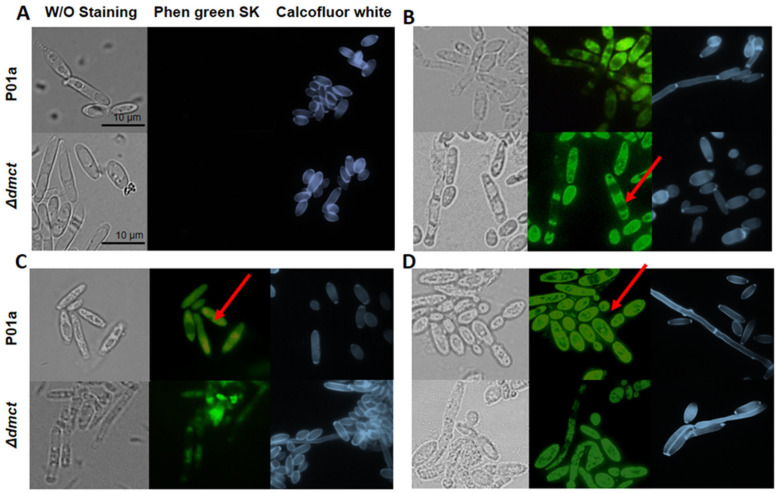
Intracellular accumulation of divalent metal cations and cell wall integrity of *Yarrowia lipolytica* P01a and Δ*dmct* strains. Cells were cultured for 20 h in YNB medium (**A**) without addition of divalent cations or (**B**) added with 16 mM zinc sulfate, (**C**) 4 mM copper sulfate, or (**D**) 4 mM iron sulfate. Cells were stained, or not, with Phen green SK or Calcofluor white (as indicated), then observed and photographed with a Leica DMRE microscope equipped with fluorescence. Red arrows indicate the areas of greatest accumulation of metals. Representative images are shown.

**Table 1 jof-09-00600-t001:** Comparative analysis of sequences * corresponding to possible structural orthologous-homologous proteins encoded by the *Yarrowia lipolytica DMCT* gene.

Protein *	Function	Organism	Length (Aa)	% Complete Gene Similarity	% CDF Domain Similarity
*YlDmct*	Hypothetical transport of metal divalent cations (This studio)	*Y. lipolytica*	555	100	100
Cot1	Transport of metal cations in vacuole, resistance to cobalt and rhodium, response to stress	*S. cerevisiae*	439	22.67	18.67
Zcr1	Resistance to zinc and cadmium	*S. cerevisiae*	442	21.07	19.41
Mmt1p	Zinc transport	*S. cerevisiae*	510	19.28	23.44
Zrg17	Zinc transport in endoplasmic reticulum	*Z. rouxii*	634	21.14	24.24
Msc2	Zinc homeostasis, zinc influx to the endoplasmic reticulum	*S. cerevisiae*	725	19.12	19.88
Mitochondrial metal transporter	Iron accumulation in mitochondria	*S. cerevisiae*	510	19.85	23.44
Cobalt toxicity protein	Zinc vacuolar transport and cobalt and rhodium resistance	*S. cerevisiae*	439	19.01	18.67
Cation diffusion facilitator family transporter	Iron, copper, zinc and cadmium transport	*C. albicans*	616	44.30	56.77
Metal cation transporter	Copper, zinc and cadmium ion efflux	*C. albicans*	626	20.73	20.56
Zinc/cadmium resistance protein	Copper, zinc and cadmium efflux	*L. elongisporus*	474	23.20	20.70
Mitochondrial protein with role in iron accumulation	Iron accumulation	*S. stipitis*	430	21.73	22.73
Zinc/cadmium resistance protein	Copper, zinc and cadmium efflux	*K. phaffii*	459	19.93	19.76

* Sequences and data corresponding to the different proteins were obtained from local alignments in pairs. Amino acid sequence length: *Y. lipolytica*, *Yarrowia lipolytica*; *S. cerevisiae*, *Saccharomyces cerevisiae*; *Z. rouxii*, *Zygosaccharomyces rouxii*; *C. albicans*, *Candida albicans*; *L. elongisporus*, *Lodderomyces elongisporus*; *S. stipitis*, *Scheffersomyces stipitis*; *K. phaffii*, and *Komagataella phaffii*.

**Table 2 jof-09-00600-t002:** Comparative analysis of sequences corresponding to possible functional homologous analogues of the putative protein encoded by the *Yarrowia lipolytica DMCT* gene.

Homologous	Cation	Organism	Location	Accesion Number *	Aa **	Identity (%)
Zrt3	Fe^2+^/Zn^2+^	*S. c.*	Vacuole/ Lysosome	P34240	503	13
Fth1	Fe^2+^	*S. c.*		P38310	465	14
		*C. g.*		Q6FJK8	435	15
Fet5	Fe^2+^	*S. c.*		P43561	622	14
		*C. g.*		A0A0W0DZF1	621	13
Cot1	Fe^2+^/Zn^2+^	*S. c.*		P32798	439	16
Zrc1	Fe^2+^/Zn^2+^	*S. c.*		P20107	442	13
Smf3	Fe^2+^	*S. c.*		Q12078	473	15
		*C. a.*		Q5ACZ8	514	14
Ctr2	Cu^2+^	*S. c.*		P38865	189	9
Ccc2	Fe^2+^/Cu^2+^	*S. c.*	Golgi apparatus	P38995	1004	13
		*C. a.*		Q5AG51	1204	11
Hmx1	Fe^2+^	*S. c.*		P32339	317	13
Fet3	Cu^2+^	*S. c.*		P38993	636	14
		*C. g.*		Q96WT3	635	15

* Accession number corresponding to the Uniprot or NCBI database. ** Amino acid sequence length: *S. c.*, *Saccharomyces cerevisiae*; *C. a.*, *Candida albicans*; *C. g.*, *Candida glabrata*.

**Table 3 jof-09-00600-t003:** Possible transcription factors (TF) with binding sites * in the promoter region of *Yarrowia lipolytica DMCT* gene.

Trancription Factor	Binding Site	Function	Genes That Regulate	Functional Homologous Predicted in *Y. lipolytica*	Reference
Stb5p	−679, −531, −407, −389, −256, −103	Involved in the transcription of transmembrane transporters.	*ZRT3*	YALI0A10637p/Q6CHB0 YALI0C15202p/Q6CBV4 YALI0F03630p/B5RSK6 YALI0F16599p/Q6C1G1 YALI0D12628p/Q6C9A9 YALI0C22990p/Q6CB01 YALI0B06853p/Q6CFH8	[57]
Aft2p	−961	In the absence of iron, which is responsible for the transcription of genes, products mobilize copper.	*FTH1*, *FET5*, *FET3*, *SMF3*	No significant similarity found in genome	[58]
Yap1p	−882	Related to cadmium tolerance, is involved in the transcription of genes that respond to ionic detoxification, besides the expression of ferredoxin and ferredoxin reductase proteins.	*ZRT1*, *FTH1*, *FET5*, *FET3*, *COT1*, *ZRC1*, *SMF3*. *CTR2*, *CCC2*, *HMX1*	YALI0F03388p/Q6C317 YALI0D09757p/Q6C9N3 YALI0B13200p/Q6CET1 YALI0F27445p/Q6C065	[59]
Cup2p	−919	Activates the transcription of metallothionein proteins when the cell is in an environment that presents excess metals.	*FET3*	CRF1/P45815 YALI0E31669p/Q6C3W4	[60]
Nrg1p	−791, −773, −738, −729, −728, −718, −647, −646, −605, −604, −588, −559, −349, −348, −81	Related to the response to stress due to pH changes	*FET3*	YALI0C12364p/Q6CC55 YALI0E07942p/Q6C6N4 YALI0D23749p/Q6C809 YALI0D18678p/Q6C8L5 YALI0B21582p/F2Z5Y0 YALI0A16841p/Q6CGR7 YALI0C05995p/Q6CCW4 YALI0F21923p/Q6C0T4 YALI0F22649p/Q6C0Q3	[61]
Skn7p	−531, −27	Responsible for regulating genes involved in the antioxidant cellular response.	*ZRT3*, *HMX1*	YALI0D14520p/Q6C937 YALI0D04785p/Q6CA95 YALI0C21340p/Q6CB75 YALI0E13948p/Q6C5Z0	[62]
Msn2p	−854, −791, −773, −738, −729, −718, −646, −604, −560, −559, −348, −300, −104, −81,	Responsible for regulating genes involved in the antioxidant cellular response.	*FET3*, *ZRC1*, *CTR2*, *CCC2*, *HMX1*	No significant similarity found in genome	[63]
Msn4p	−791, −773, −738, −729, −718, −646, −604, −559, −348, −81	Responsible for regulating genes involved in the antioxidant cellular response.	*FET3*, *CCC2*, *HMX1*	YALI0C13750p/Q6CC08	[63]

* Binding sites were identified using Yeastract software, based on homology with transcription factors from *Saccharomyces cerevisiae.*

**Table 4 jof-09-00600-t004:** Genes from *Yarrowia lipolytica* with putative participation in the transport of metals as a possible compensatory response to *DMCT* gene deletion.

Chr	Gene (Locus Tag/ID)	Similar to	Possible Function	Functional Domains
A	*YALI_A11605g/Q6CH77*	MSF transporter	Transport of small molecules by the gradient	MSF superfamily
	*YALI_A14883g/Q6CGY6*	Ferrichrome-type siderophore transporter	Iron homeostasis	MSF superfamily
B	*YALI0_B06094g/Q6CFK7*	RdgB Ca^2+^ transporter	Metal binding and calcium transport	DDHD superfamily
	*YALI0_B17864g/Q6CE78*	High-affinity potassium absorption transporter	Potassium transport	2a38euk superfamily
	*YALI0_B19250g/Q6CE20*	Ferrichrome-type siderophore transporter	Iron homeostasis	MSF superfamily
C	*YALI0_C02541g/Q6CD98*	MSF transporter	Unknown specificity	MSF superfamily
	*YALI0_C04411g/Q6CD22*	SMF1/E protein	Divalent and trivalent metal transporter	SLC5-6-like_sbd superfamily
	*YALI0_C06105g/Q6CCV9*	MSF transporter	Unknown specificity	MSF superfamily
	*YALI0_C09823g/Q6CCF5*	MSF transporter	Anion-cation simporter	MSF superfamily
	*YALI0_C10311g/Q6CCD6*	High-affinity potassium transporter	Potassium transport	K-Trans superfamily
	*YALI0_C10670g/Q6CCC4*	MSF transporter	Unknown specificity	MSF superfamily
	*YALI0_C12254g/Q6CC58*	Iron transporter in mitochondria MMT2	Transport of divalent cation metals	SelP_N superfamily, FieF domain
	*YALI0_C16225g/Q6CBR6*	MSF transporter	Unknown specificity	MSF superfamily
	*YALI0_C17105g/Q6CBM9*	MSF transporter	Unknown specificity	MSF superfamily
	*YALI0_C18051g/Q6CBK3*	YCFI metal resistance protein	Vacuolar transport with increased tolerance to metals	MRP_assoc_pro superfamily
D	*YALI0_D00319g/Q6CAS6*	Divalent cation transporter ALR1	Divalent cation transport	Alr1p-like superfamily
	*YALI0_D20064g/Q6C8F6*	MSF transporter	Unknown specificity	MSF superfamily
	*YALI0_D24651g/Q6C7X0*	MSF transporter	Unknown specificity	MSF superfamily
	*YALI0_D26818g/Q6C7M8*	SMF2 carrier protein	Manganese transport	Nramp domain
E	*YALI0_E00748g/Q6C7H7*	Zrt2 transporter	Zinc transporter	Zip superfamily
	*YALI0_E00462g/Q6C7J0*	Alr1 transporter	Divalent cations transport	Alr1p-like superfamily
	*YALI0_E11473g/Q6C691*	HAK1 transporter	Potassium transport	K-Trans superfamily
	*YALI0_E14234g/Q6C5X7*	YBT1 transporter	Involved with Ca^2+^ and metal resistance	MRP_assoc_pro superfamily
F	*YALI0_F19118g/Q6C155*	Ferrichrome-type siderophore transporter	Iron homeostasis	MSF superfamily
	*YALI0_F20922g/Q6C0X7*	Ferrichrome-type siderophore transporter	Iron homeostasis	MSF superfamily
	*YALI0_F27709g/Q6C053*	Ferrichrome-type siderophore transporter	Iron homeostasis	MSF superfamily
	*YALI0_F29711g/Q6BZZ0*	MSF transporter	Unknown specificity	MSF superfamily

Chr., chromosome; gene locus tag/ID, obtained from NCBI and Uniprot, respectively, correspond to their localization in the chromosome. Similarity and possible function were obtained by homology and similarity analysis with other proteins. Functional domains: Conserved Domains Database (CDD). The genes and the corresponding proteins were obtained from the *Y. lipolytica* CLIB122 genome available from NCBI and confirmed bioinformatically by analyzing their functional domains in the Conserved Domains Database.

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
