# Peer review of "Identification and Characterization of Dmct: A Cation Transporter in *Yarrowia lipolytica* Involved in Metal Tolerance"

_jof, 2023, doi:10.3390/jof9060600_

Round 1
Reviewer 1 Report
The paper entitled, 'Identification and Characterization of Dmct: A Cation Transporter Gene in Yarrowia lipolytica Involved in Metal Tolerance. 'provides revealed a characteristic functional domain of the cation efflux protein family and which related to divalent metal cations tolerance. And, describe and compare their physiological absence effect-growth and morphology-between the Δdmct mutant strain of Y. lipolytica-generated by deletion of the gene-, the retromutant strain-constructed by insertion of the DMCT gene- and the strain parental, under different culture conditions. However, there are still some deficiencies in this work that need to be further perfected.
1, The Three-dimensional modeling method of the YlDmct protein should be given in Materials and method section, and the credibility of this model needs to be clarified.
2, The writing format of references needs to be further unified.
3, English and manuscript writing must be significantly improved by the native speaker.
Author Response
Prof. Dr. David S. Perlin
Editor in Chief
Journal of Fungi
Attn:
Mr. Hank Zhang
Assistant Editor, JoF
Prof. Dr. Jorge L. Folch-Mallol
Guest Editor of special issue: "Extremophile Fungi: An Arising Novel Field in Perspective"
Manuscript ID: jof-2291233
Type of manuscript: Article
Title: Identification and Characterization of Dmct: A Cation Transporter Gene
in Yarrowia lipolytica Involved in Metal Tolerance, by González-Lozano et al.
We appreciate all observations made by reviewers #1, #2 and #3, and confirmed by the Assistant editor, Mr. Hank Zhang. We have proceeded to make the corrections as suggested, and we think that our manuscript has been improved by the incorporation of all his/her suggestions and/or corrections.
With respect to comments of reviewing #1:
- The Three-dimensional modelling method of the YlDmct protein should be given in Material and methods section, and the credibility of this model needs to be clarified.
R: We agree. Next paragraph was incorporated in Material and Methods Section corresponding to this item (lines 141-145): “The prediction of the three-dimensional structure of Yl-Dmct was performed using the ChimeraX software (https://www.cgl.ucsf.edu/chimerax/download.html) with the Al-phaFold and ColabFold tools [49,50]. Confidence levels were determined according to pLDDT metrics. The Predicted Align Error (PAE) was determined with Error Plot.”
- The writing format of references needs to be further unified.
R: Thank for your observation. Reference list was modified and corrected, in agreement to Instructions for Authors, by JoF.
- English and manuscript writing must be significantly improved by the native speaker.
R: The manuscript was edited by the Language Editing Services - MDPI. We send the corresponding certificated.

Reviewer 2 Report
PDF file enclosed.
The authors report in silico analysis of the Yarrowia lipolytica putative transporter Yl-Dmct and analysis of deletion strain towards different divalent cation stress, such as calcium, copper, iron and zinc. The authors investigate the changes in growth parameters as well as in dimorphic transition and ion compartmentalization. Interesting bioinformatics analysis. However, the study lacks several experimental biochemical and cellular evidence in order to demonstrate the precise metal ion transported by Dmct. It is important to determinate exact cation substrate of DMCT gene product, instead of using generalized conclusion about ion homeostasis, as the authors claim functional characterization of the Dmct protein.
Comments:
1. Abstract line 22: “Transmembrane proteins control intracellular cation concentration” – should correct since there are many transmembrane proteins that do not function as transporters mediating ion influx or efflux! Change to ” Membrane transport proteins control intracellular cation concentration.”
2. Introduction line 42: “they are part of a big cation efflux protein Family” – how big is this Family – please provide some information using ionome and transportome studies.
3. Line 58 “P-type ATPase family that catalyzes the absorption and mobilization” –what is meant by “mobilization” in the case? - P-type ATPases mediate TRANSPORT at expense of ATP hydrolysis.
4. Line 83 – please indicate which organism encode Zrg17 transporter.
5. Line 86 ‘Their location is a controversial issue because it is not conserved in the different sub-86 groups -” there is no controversy, some transporters belong to cell membrane and some are expressed in specific compartments such as vacuole, Golgi, ER.
6. Results line 208: it is not clear whether YALI0F19734p) encoded by the Y. lipolytica DMCT gene is the unique gene in Y. lipolytica genome or there are other homologous sequences – the authors should provide this information.
7. The authors used high metal concentrations in the growth medium. The authors should provide explanation for using these concentrations. It might be indicative of metal complexation, so the real free metal ion concentrations in the growth medium is not known in this case.
For instance, commonly 100 to 200 mM CaCl2 is used for both budding and fission yeast assays. Zinc and cadmium and nickel are typically tested in micromolar range. Did authors test more physiological concentrations or excess of copper (2 or 25 μmol/l CuSO4?
8. The results provided in Fig 4 clearly demonstrate that concentrations lower than 2 mM “copper” should be used, and the results provided in Fig 5C clearly demonstrate that concentrations lower than 2mM “iron” should be used. Also the author should specify exact metal salt used in the study (chloride or sulphate…) in Figure 4, 5 and in the text.
9. Figure 6 shows wild type strain grown in the presence of high calcium exhibiting nice filaments (magnification 10x), I wonder why cells are not in hyphae form (100x)? ells on iron plates appear quite sick. Scale bars should be provided for all images.
10. The results provided testing different metals are not conclusive. The authors have to investigate also the effects of metal starvation using specific chelators and provide the results using EDTA, EGTA, or ferrozine.
11. Cellular localization of Dmct must be verified by immunolocalization studies or using specific fluorescence marker tagged Dmct protein.
12. Line 190 and results: Phen green SK is used to detect a widebroad range of ions and is not specific. Precise determination using High-Resolution Mass Spectrometry Analysis should be provided or Electron Spectroscopic Imaging would be quite interesting to include in this study to characterize sequestration capacity.
13. Table 1 ; please include and provide information (identity % ) on S. pombe Zhf transporter which is located to ER membranes (Clemens et al 2002).
14. Table 3 provides very interesting analysis of putative transcription factors binding sites in DMCT gene promoter. The authors must verify the corresponding homologous TFs in Y. lipolytica genome and include the respective Y. lipolytica TFs gene names in new column. See also Garcia et al (2002) for Y. lipolytica transcription factor.
15. Table 4 – apart form NCBI accession #, the author should provide and include the gene name using Uniprot nomenclature; like they use for DMCT gene (YALI0_F19734g) or like YALI0c04411g used for Smf1 plasma membrane transporter (Cogo et al Suppl. Table S1).
16. Figure 7 – I wonder what the authors mean by “areas of accumulation of fluorescence suggesting specific sites of copper accumulation” - what kind of intracellular compartmentalization is responsible for copper sequestration?

Author Response
Prof. Dr. David S. Perlin
Editor in Chief
Journal of Fungi
Attn:
Mr. Hank Zhang
Assistant Editor, JoF
Prof. Dr. Jorge L. Folch-Mallol
Guest Editor of special issue: "Extremophile Fungi: An Arising Novel Field in Perspective"
Manuscript ID: jof-2291233
Type of manuscript: Article
Title: Identification and Characterization of Dmct: A Cation Transporter Gene
in Yarrowia lipolytica Involved in Metal Tolerance, by González-Lozano et al.
We appreciate all observations made by reviewers #1, #2 and #3, and confirmed by the Assistant editor, Mr. Hank Zhang. We have proceeded to make the corrections as suggested, and we think that our manuscript has been improved by the incorporation of all his/her suggestions and/or corrections.
With respect to comments of reviewing #2:
- Abstract, line 22: “Transmembrane proteins control intracellular cation concentration” – should correct since there are many transmembrane proteins that do not function as transporters mediating ion influx or efflux! Change to ” Membrane transport proteins control intracellular cation concentration
R: According to the recommendation, sentence in abstract (line 22) was modified “Membrane transport proteins control intracellular cation concentration.”
- Introduction line 42: “they are part of a big cation efflux protein Family” – how big is this Family – please provide some information using ionome and transportome studies.
R: We agree. The sentence was modified and next paragraph was included (lines 41-53) in the Introduction section: “First described in 1995 by Nies and Silver in archaea, bacteria, and eukaryotes, these transporters are part of a large cation efflux protein family responsible for the mobilization of divalent metal cations such as Zn2+, Cd2+, Co2+, Fe2+, Ni2+, Mn2+, and possibly Cu2+ and Pb2+ [8-10]. In 2007, Montanini et al. proposed the subdivision of cation efflux protein fam-ily members into three main groups based on their specificity for the transported metal: Group 1, Mn-CDF; Group 2, Fe/Zn CDF with the Fe2+ and Zn2+ substrates and other metal ions; and Group 3, Zn-CDF with Zn2+ substrates and other metal ions, but not including Fe2+ or Mn2+ [11]. Later, in 2013, Cubillas et al. refined the accuracy of this classification and suggested a new division, separating the CDF proteins into 18 independent clades based on the specificity of the transported metal [12]. Recently, Xu et al. identified the first CDF protein that, in addition to mobilizing Zn2+ ions, mobilizes Na+ ions, proposing the incorporation of a new CDF group: Na-CDF [13]”.
- Line 58 “P-type ATPase family that catalyzes the absorption and mobilization” –what is meant by “mobilization” in the case? - P-type ATPases mediate TRANSPORT at expense of ATP hydrolysis.
R: According to the recommendation, line 58 (now, line 71-73) was modified: “the P-type ATPase family mediate the transport of Cd+, Cu+, H+, K+, Na+, Mg+, and Ca+ ions at expense of ATP- hydrolysis”
- Line 83 – please indicate which organism encode Zrg17 transporter.
R: We agree. The sentence (now in line 98) was modified: “for example, the Zrg17 protein from S. cerevisiae which is located in ER… “
- Line 86 ‘Their location is a controversial issue because it is not conserved in the different sub-groups -” there is no controversy, some transporters belong to cell membrane and some are expressed in specific compartments such as vacuole, Golgi, ER.
R: We agree. Thank for the recommendation; the sentence (now in lines 101-103) was restructured: “CDF proteins are located in different membrane types, such as the cell membrane, and some are expressed in membranes belonging to specific compartments, such as the vacuole, Golgi, and ER [10,27]”.
- Results line 208: it is not clear whether YALI0F19734p) encoded by the Y. lipolytica DMCT gene is the unique gene in Y. lipolytica genome or there are other homologous sequences – the authors should provide this information.
R: In attention to this recommendation, we add (lines 236-237, in the new version of manuscript) the next paragraph: “According to our comparative genomics analysis, in the genome of Y. lipolytica, there is only one copy of the DMCT gene, and no homologues were found”.
- The authors used high metal concentrations in the growth medium. The authors should provide explanation for using these concentrations. It might be indicative of metal complexation, so the real free metal ion concentrations in the growth medium is not known in this case. For instance, commonly 100 to 200 mM CaCl2 is used for both budding and fission yeast assays. Zinc and cadmium and nickel are typically tested in micromolar range. Did authors test more physiological concentrations or excess of copper (2 or 25 μmol/l CuSO4?
R: Thanks for your comment. Initially, we agreed with what you mentioned, and we tested the cation concentrations reported in the literature; however, the physiological development of the strains did not show changes in the analysed parameters, so we decided to work it at higher concentrations as it is reported by some authors who work towards the extremophility of yeasts (Li et al. 2013; Siloniz et al 2002). We analyze yeast morpho/physiology at concentrations of calcium from 10 mM to 1M, copper from 1 to 10 mM, Iron 1 to 10 mM, zinc from 2 to 10 mM, manganese 1 to 30 mM, and mangenium from 1 to 30 mM; additionally, we proved the absence of these metals using the Synthetic Defined medium (previously reported: Martha-Paz et al. (2019) and Amberget al.,2005). without metals added. Under all tested conditions, we observed growth of all the strains. For the present manuscript, we decided to show the results corresponding only to the conditions where we obtained signs of a different phenotype.
- The results provided in Fig 4 clearly demonstrate that concentrations lower than 2 mM “copper” should be used, and the results provided in Fig 5C clearly demonstrate that concentrations lower than 2mM “iron” should be used. Also the author should specify exact metal salt used in the study (chloride or sulphate…) in Figure 4, 5 and in the text.
R: We think that the response to question #7 could be explaining, in part, this question. Also, in this type of experiments (corresponding to Fig. 4) the goal is to show that concentration at which a cellular response (in this case, the ability to grow or not) becomes visible when the cell is exposed to the presence of different concentrations of the agent to be tested.
We believe that figure 4 clearly demonstrates the concentration of each of the agents (ions tested), which is capable of generating changes in the growth of Y. lipolytica cells, in each of the strains analysed. Also, including all the concentrations tested leads to generating a very large figure from which, in many of the panels, no differences in the genotype of the cells will be observed.
Concerning to the salt type, in the Materials and methods section (lines 214-216), we added the next sentence: “For the phenotypic characterization, in all the experimental series, divalent cations were used as chloride salts (or iron sulfate, when is indicated), which were obtained from Jalmek, México.”
- Figure 6 shows wild type strain grown in the presence of high calcium exhibiting nice filaments (magnification 10x), I wonder why cells are not in hyphae form (100x)? cells on iron plates appear quite sick. Scale bars should be provided for all images.
R: Figure 6 shows the morphology of the Y. lipolytica strains in the presence of calcium, copper, iron and zinc ions in solid YNB medium. In this figure we treat to show the effect that the presence of different cations has on the cell morphology, and to compare such effect between parental and mutant strains. The main differences that you highlight could be due to the following: colonies (outer boxes, 10X) correspond to cultures in YNB solid medium for 36 h, while for the microscopic observations (inner boxes, 100X) they were carried out at 20h in YNB liquid medium. In the legend of Fig 6, it is explained the condition corresponding to each image.
Scale bars should be provided…
We understand that this question is optional. In the scientific literature there are different forms to show this type of figures (photography); as an example, please, see J. Fungi 2021, 7, 463. https://doi.org/10.3390/jof7060463 (Fig: 6A, 7A, 7B, 9, 10E, 14A, 14B and 14C…). However, thanks for this observation, we added the value to the scale bar, in each photography.
- The results provided testing different metals are not conclusive. The authors have to investigate also the effects of metal starvation using specific chelators and provide the results using EDTA, EGTA, or ferrozine.
R: Thanks for your comment. As you proposed we tested metal starvation with EDTA, but we do not observed changes in the cell phonotype; due that, we decided not to show the results with this chelating agent. Also, we did experiments using Synthetic Defined medium (SD) without cations added. SD medium has been reported previously by Amberg et al.,2005, and Martha-Paz et al., 2019; doi.org/10.1080/09687688.2019.1667034.
- Cellular localization of Dmct must be verified by immunolocalization studies or using specific fluorescence marker tagged Dmct protein.
R: Thank you, this is an excellent idea. At this stage of this work, our objective is to present the results obtained so far, corresponding to the description of the gene product and its possible function in the cell, in response to different concentrations of cations.
We will consider it for next stage in this work. Additionally, we think too make the expression analysis both DMCT gene (under different conditions and concentration of cations) as genes corresponding to the putative transcriptions factors identified as possible regulators of DMCT gene (mentioned in table 3).
Unfortunately, at this moment it is not possible for us, to make these interesting analyses.
- Line 190 and results: Phen green SK is used to detect a widebroad range of ions and is not specific. Precise determination using High-Resolution Mass Spectrometry Analysis should be provided or Electron Spectroscopic Imaging would be quite interesting to include in this study to characterize sequestration capacity.
R: As you rightly comment, the Phen green™ SK fluorescent heavy metal indicator, can be used to detect a wide range of ions, including Cu2+, Cu+, Fe2+, Hg2+, Pb2+, Cd2+, Zn2+, and Ni2+. It has been widely used by various authors for quantitative studies of iron uptake, characterization of ion selectivity by transmembrane transporters and for monitoring the flow of metals such as copper through organelle membranes. Our objective with this was solely to establish a comparison in the patterns of ions accumulation at the subcellular level between the parental (with DMCT gene) and mutant (without DMCT gene) strains.
At this moment, it is not possible for us, to make the interesting analyses suggested.
- Table 1; please include and provide information (identity %) on S. pombe Zhf transporter which is located to ER membranes (Clemens et al 2002).
R: Table 1 shows the summary of the multiple alignment analysis of the putative protein encoded by the DMCT gene, which allowed us to identify regions that present similarities between this protein and possible homologous proteins in other species. For the selection of possible homologous proteins, we considered different parameters highlighting the architecture of the functional domains, the conservation of some amino acid residues, the length of the sequences and their percentages of similarity both of complete protein and its conserved CDF domain. When we performed the relevant analyses with the S. pombe Zhf (Zinc ion transporter) protein, some difficulties arose: 1) The protein encoded by the DMCT gene has 555 amino acid residues, whereas Zhf only has 387 amino acid residues. 2) The similar amino acid residues are extremely few; when comparing both sequences we obtained a 22.9% similarity but with a coverage of 6.68%, so we consider that is not conclusive information to consider to Zhf as a homologous protein of YlDmct.
- Table 3 provides very interesting analysis of putative transcription factors binding sites in DMCT gene promoter. The authors must verify the corresponding homologous TFs in Y. lipolytica genome and include the respective Y. lipolytica TFs gene names in new column. See also Garcia et al (2002) for Y. lipolytica transcription factor.
R: We appreciate your important recommendation. We have already added this information in table 3.
- Table 4 – apart form NCBI accession #, the author should provide and include the gene name using Uniprot nomenclature; like they use for DMCT gene (YALI0_F19734g) or like YALI0c04411g used for Smf1 plasma membrane transporter (Cogo et al Suppl. Table S1).
R: Thank you for your comment. We have added the NCBI systemic name and UniProt nomenclature in Tables 3 and 4.
- Figure 7 – I wonder what the authors mean by “areas of accumulation of fluorescence suggesting specific sites of copper accumulation” - what kind of intracellular compartmentalization is responsible for copper sequestration?
R: We agree with this comment. The sentence was changed to “in contrast, in the mutant cells (Δdmct), there were areas of accumulation of fluorescence, suggesting copper accumulation in cells lacking the DMCT gene”.
Reviewer 3 Report
I have revised the manuscript, and here are the comments:-
The authors need to check the language of the manuscript.
Many abbreviations in the manuscript need to be cleared, and add more results in the abstract.
Please arrange all keywords in alphabetical order
Provide the origin and model of all devices.
In addition, many scientific names in the manuscript must be written in italic format.
To improve the quality of the paper, update the reference list by adding 2022 and 2023 references.
Please follow the authors' instructions on how they write the reference in the list. Why are you capitalizing the first letter of every word? Please see the journal style. For references about textbooks, please add the page numbers of the textbook. Also, please add the city of the publisher.
Please make all tables self-explanatory and do not use abbreviations in the table footnote or the table legend.
Enhance the resolution of all figures.
The conclusion shouldn’t contain results, check and rewrite this part.
Regarding the tables: Why did you put the tables in the form of an image in the manuscript; please provide original data, and Clear the novelty and hypothesis of the manuscript.
Author Response
Prof. Dr. David S. Perlin
Editor in Chief
Journal of Fungi
Attn:
Mr. Hank Zhang
Assistant Editor, JoF
Prof. Dr. Jorge L. Folch-Mallol
Guest Editor of special issue: "Extremophile Fungi: An Arising Novel Field in Perspective"
Manuscript ID: jof-2291233
Type of manuscript: Article
Title: Identification and Characterization of Dmct: A Cation Transporter Gene
in Yarrowia lipolytica Involved in Metal Tolerance, by González-Lozano et al.
We appreciate all observations made by reviewers #1, #2 and #3, and confirmed by the Assistant editor, Mr. Hank Zhang. We have proceeded to make the corrections as suggested, and we think that our manuscript has been improved by the incorporation of all his/her suggestions and/or corrections.
With respect to comments of reviewing #3:
We answered and corrected all the issues mentioned by the reviewer as it follows:
- The authors need to check the language of the manuscript.
R: The manuscript was edited by the Language Editing Services - MDPI. We send the corresponding certificated.
- Many abbreviations in the manuscript need to be cleared, and add more results in the abstract.
R: We agree. Abbreviations in the document were clarified.
… add more results in the abstract..
R: We do not agree. The Journal limit the number of words in the abstract to 200 words. From 199 words included in our abstract, 136 (68.3%) correspond to results obtained in this work.
- Please arrange all keywords in alphabetical order: cation efflux; divalent metal cation transporter; DMCT gene; Yarrowia lipolytica; YALI0F19734p; YALI_F19734g; Yl-Dmct hypothetical protein.
R: Thank you for your recommendation. The keywords were ordered in alphabetical order (lines 32-33).
- Provide the origin and model of all devices.
R: We agree. The recommendation was attended.
- In addition, many scientific names in the manuscript must be written in italic format.
line 97 Yarrowia; line 475 in silico; lines 524-525 Geotrichum, Clavispora, Debaryomyces, Pichia, Ogataea; Line 528 Debaryomyces; Line 560 Yarrowia
R: All of them were corrected.
… lines 499, 501, 505, 509, 566, 601, 610, 631, 731, 757, 776 et al.
R: We agree with this observation; notwithstanding, the format in this journal do not consider the use of italic in this latin locution (et al.,)… Please, see: J. Fungi 2023, 9(3), 308; https://doi.org/10.3390/jof9030308).
- To improve the quality of the paper, update the reference list by adding 2022 and 2023 references.
The complete document has been modified; also, the list of references was modified (and corrected) too.
- Please follow the authors' instructions on how they write the reference in the list. Why are you capitalizing the first letter of every word? Please see the journal style. For references about textbooks, please add the page numbers of the textbook. Also, please add the city of the publisher.
R: Thank for your recommendation. The complete list of references was corrected.
- Please make all tables self-explanatory and do not use abbreviations in the table footnote or the table legend.
Linea 239 Yarrowia lipolytica; Linea 253 Yarrowia lipolytica; Linea 319 Yarrowia lipolytica;Linea 322 transcription factors Saccharomyces cerevisiae; Linea 480 Yarrowia lipolytica.
R: Thank for your recommendation. All footnote and/or table legend were modified.
- Enhance the resolution of all figures.
R: We think this will be a later step in the publishing process.
- The conclusion shouldn’t contain results, check and rewrite this part.
R: Thank for your recommendation. The complete document was edited and modified, and we do not can identify results in the conclusion.
- Regarding the tables: Why did you put the tables in the form of an image in the manuscript; please provide original data,
R: I understand that the format of J of F, it is only to facility the revision of manuscripts by reviewers and they can see all information in one document. Subsequently, if the manuscript is acceptable or accepted, the Editor Manager will request the images (tables and figures) in format(s) that ensure the necessary quality for its publication.
- Clear the novelty and hypothesis of the manuscript.
R: We agree. We added the next paragraph in the Conclusions section:
“The results presented here suggest that, as mentioned repeatedly the DMCT gene (YALI0F19734g) from Y. lipolytica encoded a protein which characteristics permit to consider it as a member of the Cation Difussion Facilitador (CDF) family. Furthermore, this represents the first report about the characterization of a member of CDF family in this specie of great biotechnological importance”.

Round 2
Reviewer 2 Report
The authors have satisfactorily addressed most of my comments.
The manuscript has been revised according to the comments and suggestions and has been improved.
The authors will address the main concern regarding YlDMCT localization and precise substrate(s) in future studies.
Please see minor corrections prior to acceptance:
Line 68: correct K(2+) to K(+)
Line 104: correct “the protein encoded by the ZRC1 and COT1 genes has been reported in vacuoles” to “the proteins encoded by the ZRC1 and COT1 genes have been reported in vacuoles” and cite MacDiarmid CW, Gaither LA, Eide D. Zinc transporters that regulate vacuolar zinc storage in Saccharomyces cerevisiae. EMBO J. 2000 Jun 15;19(12):2845-55. doi: 10.1093/emboj/19.12.2845.
Author Response
Please see minor corrections prior to acceptance:
Line 68: correct K(2+) to K(+)
R: Thank you for this observation; the error was corrected.
Line 104: correct “the protein encoded by the ZRC1 and COT1 genes has been reported in vacuoles” to “the proteins encoded by the ZRC1 and COT1 genes have been reported in vacuoles” and cite MacDiarmid CW, Gaither LA, Eide D. Zinc transporters that regulate vacuolar zinc storage in Saccharomyces cerevisiae. EMBO J. 2000 Jun 15;19(12):2845-55. doi: 10.1093/emboj/19.12.2845.
R: Thank you for your observation; the sentence was corrected and the reference was incorporated.
Following his/her previous comments, in the manuscript, the references were rearranged and some were corrected (format). Table 4 was modified with new numbers in the cited references. Also, the tables were incorporated in their original format.
Reviewer 3 Report
The authors have carefully processed all comments. The quality of the manuscript has increased significantly. I have no further comments.
Author Response
The authors have carefully processed all comments. The quality of the manuscript has increased significantly. I have no further comments.
R: Thank you for your comment(s). In the process, the quality of manuscript, following such comments, was increased.
Following his/her previous comments, in the manuscript, the tables were incorporated in their original format. Also, the references were rearranged and some were corrected (formatting).